# *Dendrobium officinale* Enzyme Changing the Structure and Behaviors of Chitosan/γ-poly(glutamic acid) Hydrogel for Potential Skin Care

**DOI:** 10.3390/polym14102070

**Published:** 2022-05-19

**Authors:** Mengmeng Wang, Erwei Zhang, Chenrui Yu, Dandan Liu, Shiguang Zhao, Maodong Xu, Xiaofeng Zhao, Wenjin Yue, Guangjun Nie

**Affiliations:** 1College of Biological and Food Engineering, Anhui Polytechnic University, Wuhu 241000, China; wangmm@ahpu.edu.cn (M.W.); zhangerwei@ahpu.edu.cn (E.Z.); yuchenrui@ahpu.edu.cn (C.Y.); ddliu@ahpu.edu.cn (D.L.); zhaosg@ahpu.edu.cn (S.Z.); 2School of Chemical and Environmental Engineering, Anhui Polytechnic University, Wuhu 241000, China; xuemaodong@ahpu.edu.cn; 3College of Life and Environmental Sciences, Hangzhou Normal University, Hangzhou 310036, China; xiaofengzhao@yahoo.com

**Keywords:** hydrogel, chitosan, γ-poly(glutamic acid), *Dendrobium officinale* enzyme, skin care

## Abstract

Hydrogels have been widespreadly used in various fields. But weak toughness has limited their further applications. In this study, *Dendrobium officinale* enzyme (DOE) was explored to improve chitosan/γ-poly(glutamic acid) (CS/γ-PGA) hydrogel in the structure and properties. The results indicated that DOE with various sizes of ingredients can make multiple noncovalent crosslinks with the skeleton network of CS/γ-PGA, significantly changing the self-assembly of CS/γ-PGA/DOE hydrogel to form regular protuberance nanostructures, which exhibits stronger toughness and better behaviors for skin care. Particularly, 4% DOE enhanced the toughness of CS/γ-PGA/DOE hydrogel, increasing it by 116%. Meanwhile, water absorption, antioxygenation, antibacterial behavior and air permeability were increased by 39%, 97%, 27% and 52%.

## 1. Introduction

Hydrogels are hydrophilic polymers with three-dimensional networks accommodating a large amount of water, exhibiting flexibility like natural tissue. In particular, hydrogels made with natural polymers like chitosan (CS) and γ-polyglutamic acid (γ-PGA) have exhibited superior biocompatibility, biodegradability, elasticity and permeability to oxygen and nutrients [1,2,3,4]. It belongs to generally recognized as safe (GRAS) products, attracting increasing attention in the fields of medicine and tissue engineering. Most of hydrogels own uneven crosslinking networks constructed with natural macromolecules [5,6,7], which topology structures are so uneven as to lack effective energy dissipation mechanism that prevents crack propagation and stress concentration. As a non-uniform hydrogel being subjected to external forces, the stress is rapidly transferred and concentrated in the structural defects or weak crosslinking points, resulting in rapid crack propagation and hydrogel cracking, which seriously limit its practical application [8,9]. The toughness of hydrogel is mainly determined by the following points. (1) Molecular chain length: The length affects the uniformity of hydrogel network. When the inhomogeneous network is stressed by external force, stress-concentrated point easily cracks [8,9]. (2) Crosslinking way: Covalent crosslinking and noncovalent crosslinking are two main ways to form hydrogels [10]. Noncovalent crosslinking includes ion interactions, hydrogen bonds, hydrophilic interactions, hydrophobic interactions, metal coordinations and protein-specific interactions [11,12]. All of them belong to physical crosslinking, which is usually reversible and can protect the skeleton of hydrogels through energy consumption. The reversibility of the weak bond dissipating energy can improve the toughness of hydrogels. As being of chemical crosslinking, covalent crosslinking is so stronger that the structural network of hydrogel can be strengthened [13,14]. (3) Crosslinking point: Crosslinking degree depends on the number and type of crosslinking point in skeleton molecules of hydrogels. More crosslinking points mean more linking chances available for uniform network to improve their strength and toughness. Therefore, four strategies have been created to improve the toughness of hydrogels: (1) Introducing sacrifice units to dissipate energy [13,14], (2) increasing crosslinking points [15], (3) improving the uniform of hydrogel network [16], (4) introducing “molecular sliding mechanism” or stronger dynamic interaction system [17].

CS/γ-PGA hydrogel is one of many important traditional hydrogels. CS is the only natural polycation polysaccharide that can be combined with carboxyl group. γ-PGA owns abundant hydrophilic carboxyl groups. They have often been applied to construct the skeleton network of hydrogel via polyelectrolyte complexing (PEC) interactions. However, PEC interactions among macromolecules easily result in inhomogeneous network of hydrogels with low toughness. In this present work, the strategy of multiple noncovalent crosslinking based on *Dendrobium officinale* enzyme (DOE) was introduced to induce self-assembly of CS/γ-PGA hydrogel, improving the toughness (Figure 1).

*Dendrobium officinale*, the dry stem of the orchid species *Dendrobium officinale* Kimura et Migo, is one of the traditional Chinese medicinal herbs, which has wealthy medicinal efficacy and thus has been recognized as a high-quality health food in China and other Southeast Asian countries [18]. *Dendrobium* plants have lots of various *Dendrobium officinale* polysaccharides (DOPs) [19] and proteins. Microbial fermentation of *Dendrobium* plants can produce various enzymes including amylase, protease and so on, which can hydrolyze polysaccharides and proteins into low-molecular carbohydrates, peptides, and various amino acids [20]. In this present work, the fermented *Dendrobium officinale* is named as *Dendrobium officinale* enzyme (DOE), which contains various monosaccharides, polysaccharides, amino acids and peptides. They can permeate different size of pores encompassed by the chains of CS and γ-PGA and make multiple noncovalent crosslinking with CS/γ-PGA skeleton through PEC interactions, hydrogen bond, hydrophilic interactions, hydrophobic interaction, etc., enhancing the uniformity of the network required for the toughness of CS/γ-PGA hydrogel [21,22]. On the other hand, these components exhibit a few biological activities including antioxidant, anticancer, antifatigue, immunoenhancement to name a few [23,24]. *Dendrobium* plants have various anionic polysaccharides such as stilbenoid, managing melanogenesis, skin-darkening and skin-aging and more particularly. Many antioxidantive DOPs can remove reactive oxygen species and oxidative free radicals [19], regulate the balance of collagen production and degradation, and protect photoaging human skin fibroblasts [25]. With a better skin hydrating efficacy, these DOPs have been regarded as a safe and efficient protector against skin dryness [26]. Therefore, DOE can provide desired skin care. In view of the importance of skin moisturizing, nourishing, antioxidation and air permeability for skin care, the effects of γ-PGA and DOE contents on the behaviors of hydrogel were also studied in this work.

## 2. Materials and Methods

### 2.1. Materials and Chemicals

CS (average molecular weight: 370 kD, deacetylation ≥ 80%), DPPH, Coomassie blue sodium chloride, glacial acetic acid, and bovine serum albumin (BSA) were of analytically pure and purchased from Sinopharm chemical reagent co. LTD (Beijing, China). γ-PGA crystalline (average molecular weight: 700 kD, purity ≥ 95%), being of commercial standard, was produced by *Bacillus subtilis* natto. The strains of *Lactobacillus*, *Staphylococcus aureus* (*S. aureus*), and *Escherichia coli* (*E. coli*) were purchased from China Center of Industrial Culture Collection (CICC) and China General Microbiological Culture Collection Center (CGMCC), respectively.

### 2.2. Preparation of DOE

According to the previous method [20], 5 mL *Lactobacillus* solution (1.0 × 10^7^–1.0 × 10^8^ cfu/mL) was inoculated in the culture composed of 2.5 g cleaned *Dendrobium officinale* stem and 11.25 mL deionized water. The inoculated culture was anaerobically incubated at 28 °C for 30 d, and the fermented broth was centrifuged at 6000 rpm for 10 min. The supernate was defined as DOE.

### 2.3. Preparation of Precursor Solution of Hydrogel

Firstly, 100 mg γ-PGA was fully dissolved in 15 mL pH 9.0 sodium hydroxide solution, in that 300 mg CS was added and stirred at 550 rpm and 45 °C for 5 min to be uniformly distributed. After the total concentration of CS and γ-PGA being fixed as 2% (*w*/*v*), the content of γ-PGA was separately controlled as 10%, 14%, and 25% in the total concentration. Secondly, 5 mL pH 2.2 glacial acetic acid was dropwise added in CS/γ-PGA solution at one droplet per second. Thirdly, 62.5 μL original DOE solution was slowly added in the solution and stirred at 550 rpm and 45 °C for 45 min. The final solution was called as precursor solution of hydrogel.

### 2.4. Preparation of Hydrogel

Based on the reported method [27], 20 mL precursor solution of hydrogel was slowly poured into a Petri dish (*Φ* 9 cm). The hydrogel was formed by heat-drying at 40 °C overnight or freeze-drying. The dried hydrogel was immersed in 15 mL 0.1 mol/L sodium hydroxide solution for 24 h and then rinsed with deionized water until the pH value of the used deionized water unchangeable. Finally, the hydrogel was placed at 45 °C to be dry.

### 2.5. SEM, FTIR, XRD and DTG

The morphological images of the CS, γ-PGA/CS, and γ-PGA/CS/DOE hydrogels were obtained with a S-4800 SEM (Hitachi, Tokyo, Japan) at 5 kV. FTIR spectra of samples were collected at 4 cm^−1^ resolution in the wavenumber range of 400–4000 cm^−1^ using the FTIR spectrometer (Shimadzu IRPrestige-21, Kyoto, Japan). After heat-dried at 60 °C for 12 h, the samples were ground with dry potassium bromide and pressed into a tablet for collecting FTIR spectra. The crystalline nature of the samples was analyzed using the XRD instrument (Bruker, Karlsruhe, German). The scan speed was set as 2°/min in the range 10 to 40° and the X-ray tube was operated at 40 kV and 40 mA. Thermogravimetric analysis (TGA) of samples was conducted with the microcomputer differential thermal balancer (Shimadzu DTG-60H). 8–10 mg sample was heated from 30–700 °C at 10 °C/min in 20 mL/min N_2_. Derivative thermogravimetric (DTG) curves were obtained using OriginPro 9.0 by differentiating the TGA curves.

### 2.6. Adsorption of Bovine Serum Albumin

The previous method [28] was applied to evaluate the behavior of hydrogel to adsorb BSA. 100 mg of dry hydrogel with 1 cm × 1 cm size was mixed with 50 mL of 1.0 mg/mL BSA solution at room temperature for 60 h. After filtered with 0.45 μm film, this solution was assayed by UV spectroscopy at 280 nm. The adsorbed amount of BSA was calculated from a standard calibration curve. All experiments were performed in triplicate. The adsorption ratio (*Qa*) was calculated by Equation (1).
(1)$$Qa=\frac{\left(C_{0}-C_{1}\right)\times V}{S}$$
where, *C*_0_ and *C*_1_ are the concentrations (mg/mL) of BSA absorbed before and after, respectively. *V* is the volume (mL) of BSA solution. *S* is the size (m^2^) of dry hydrogel used.

### 2.7. Water Absorption and Retention

The behaviors of hydrogels in water absorption and retention were measured using the reported method [29]. The sample was cut into pieces with 1 × 1 cm^2^ size and dried at 45 °C until constant weight (*M*_0_). Then the pieces were immersed in distilled water for 60 min, hanged for 60 s and the free water on their surfaces was removed. The weight at this time was named as *M*_60_. The pieces with *M*_60_ weight were transferred in a tube with a filter, centrifuged at 3500 rpm for 3 min, and weighed as *M_d_*. Water absorption ratio (*Rwa*) and water retention ratio (*Rwr*) were calculated respectively using Equations (2) and (3).
(2)$$Rwa=\frac{M_{60}-M_{0}}{M_{0}}\times 100\%$$
(3)$$Rwr=\frac{M_{d}-M_{0}}{M_{60}-M_{0}}\times 100\%$$

### 2.8. Antibacterial Behavior

The method of filter paper-agar diffusion was employed to estimate the antibacterial behavior of hydrogels. Several sterile round papers (*Φ* 6 mm) were immersed in different precursor solution of hydrogels for 30 min. 0.1 mL *S. aureus* or *E. coli* (1.0 × 10^6^–1.0 × 10^7^ cfu/mL) was inoculated in beef paste agar medium in Petri dishes (*Φ* 9 cm). The immersed papers were orderly placed on the inoculated medium. After the medium being incubated at 37 °C for 24–48 h, the diameters of inhibition circles were assayed to determine antibacterial behavior of hydrogels.

### 2.9. Tensile Test

An electronic single yarn strength tester (Darong YG021DL, Wenzhou, China) was used for the uniaxial tensile tests. The specimens were shaped into a strip with 50 mm in length, 5 mm in width and 1.5 mm in thickness. The measuring range, the pre-tension and the clamping distance were designed as 5000 cN, 10 cN and 30 mm, respectively. The tests were carried out at a stretching rate of 30 mm·min^−1^.

### 2.10. DPPH^+^ Scavenging Behavior

DPPH^+^ radical has widely been used to evaluate the free radical scavenging ability of natural compounds [30]. The DDPH^+^ scavenging activity (*A_ds_*) was assayed to estimate antioxygenic property of hydrogels. 40 mg hydrogel with different concentration of DOE (3–7%) was grinded into powder and added in 1 mL distilled water. After mixed for 8 h, the mixture was centrifuged at 8000 rpm for 10 min. Then, 100 μL supernatant was mixed with 100 μL 0.2 mM 1,1-diphenyl-2-picrylhydrazyl radical (DPPH) solution with ethanol as solvent in a 96-well plate. After placed for 30 min in dark, the mixed solution was assayed by multifunctional microplate analyzer (Multiskan FC) at 517 nm. *A_ds_* was calculated from Equation (4).
(4)$$A_{ds}=\left[1-\frac{A_{1}-A_{2}}{A_{0}}\right]\times 100\%$$
where, *A*_0_, *A*_1_, and *A*_2_ refer to OD_517_ nm values of DPPH without sample, DPPH with sample, and sample without DPPH, respectively.

### 2.11. MTT Test

The viability of HEK-293 cell was determined by methyl thiazolyl tetrazolium (MTT) assay (Sigma-Aldrich, St. Louis, MO, USA). Briefly, cells were seeded in 96-well microplates and grown on hydrogel samples with cell culture media (100 μL/well) at a density of 1 × 10^4^ cell/mL for 24 h. 15 μg sample was washed once with the incomplete cell culture media, followed by adding 10 μL of MTT reagents and 90 μL of incomplete cell culture media to each well to allow the cells to grow for 4 h in 5% CO_2_ humidified atmosphere in a CO_2_ incubator (Therom, Waltham, MA, USA) at 37 °C, then added 110 uL Formazan. Finally, the microplates were analyzed with a microplate reader (Bio-Rad, Hercules, CA, USA), and the absorbance value at 490 nm for each well was measured. The viability of cells was expressed as the percentage of the experimental group to the control group. All experiments were repeated in three parallel repeats.

### 2.12. Contact Angle and Rolling Angle

Dry CS/γ-PGA/DOE hydrogel was shaped into 0.5 × 0.5 cm^2^ blocks, and then they were immersed in 15 mL distilled water for 5 min. The blocks were taken out to remove free water from the surface and then immobilized on clear glass slides with double-sided tape. The immobilized hydrogel samples were determined in contact angle using an optical contact angle meter (KINO SL250, Boston, MA, USA). Dry CS/γ-PGA/DOE hydrogel was shaped into 1.0 × 2.0 cm^2^ blocks, and they went through the same treatment mentioned above for assaying rolling angle.

### 2.13. Air Permeability

In this work, air permeability was represented by oxygen permeability, which was tested by isobaric method. A specific glass container with an air inlet and a connect port was customized. Before testing, the oxygen in the container was completely expelled by N_2_, and then the inlet was covered with CS/γ-PGA/DOE hydrogel. The content of oxygen in the container was measured in real time by an oxygen meter set in the connect port. Air permeability of hydrogel was evaluated using the product of diffusion coefficient *D* with solubility coefficient *k*. The value of *Dk* was calculated by Equation (5).
(5)$$Dk=\frac{V\cdot L\cdot dC\;}{S\cdot P\cdot dt}\;$$
where, *C* is the mean of oxygen content in the container (%), *P* is the oxygen partial pressure difference on both sides of the hydrogel (0.21-C = 760 mmHg), *V* is the volume of the container (mL), *S* and *L* are the area (cm^2^) and the thickness (cm) of the hydrogel to be tested, respectively. *t* (s) is the time for oxygen permeating.

### 2.14. In Vitro Release of DOE

Based on the reported method [31], CS/γ-PGA/DOE hydrogel was added in 10 mL PBS buffer solution (pH 7) and shook at 100 rpm and 37 °C for 3 h. Every a certain time, 3 mL solution was taken out (supplementing 3 mL fresh solution) to be measured by UV-spectrophotometer at 270 nm, and then the released DOE was evaluated according to Equation (6).
(6)$$Cumulative\;Release=\frac{V_{e}\sum_{1}^{n-1}C_{i}+V_{0}C_{n}}{M}\times 100\%$$
where, *V_e_* and *V*_0_ refer to is 3 mL and 10 mL. *C_n_* is the concentration of DOE in the solution after *n* times sampling, and *M* is the total mass of DOE in the hydrogel. All assays were done in three repeats.

It was found that DOE contains affluent phenols, which have a characteristic peak at 270 nm [32]. The content of DOE was assayed using ultraviolet spectrum at 270 nm.

## 3. Results and Discussion

### 3.1. Effect of γ-PGA

To investigate the effect of γ-PGA content on CS/γ-PGA hydrogel, the hydrogels with 10%, 14%, and 25% γ-PGA were studied in the behaviors of BSA adsorption, water absorption, water retention and antibiosis. As highlighted in Figure 1A, the hydrogel with a higher content of γ-PGA can adsorb more BSA. The hydrogel with 25% γ-PGA exhibited the highest adsorption capacity, 1922 mg/m^2^, followed by 1066 mg/m^2^ with 14% γ-PGA, and the last 731 mg/m^2^ with 10% γ-PGA. Probably, more γ-PGA molecules introduced more carboxyl groups as more available adsorption sites for BSA binding [33]. As illustrated in the insets of Figure 1A, the hydrogel with 25% γ-PGA obviously adsorbed BSA from Coomassie blue solution. Theoretically, the adsorption behavior correlates with the structural porosity of hydrogels. The increase of BSA adsorption may be ascribed to the rise in the porosity. As shown in Figure 2A, neat CS hydrogel has a compact structure with many random crevices. Differently, CS/γ-PGA hydrogel has a more evident texture structure with many regular and deep gullies, possibly resulting from PEC interactions between amino groups of CS and carboxyl groups of γ-PGA [34,35]. Reasonably, these gullies can improve the adsorption behavior of CS/γ-PGA hydrogel due to capillary siphoning. It has been speculated that γ-PGA with enormous carboxyl groups could change the structures of CS/γ-PGA hydrogel so as to improve the hydrophilicity. Finally, the behaviors of water absorption and retention were further influenced. As proved in Figure 1B, the behaviors of water absorption and retention of CS/γ-PGA hydrogel both increased with γ-PGA. To our best knowledge, the size of pores and the intermolecular spaces inside hydrogel network can remarkably adjust the behaviors [36]. With the increase of γ-PGA content, more PEC interactions possibly take place between CS and γ-PGA. The introduced γ-PGA chains separate the chains of CS from each other, creating more spaces to accommodate much more water. The higher porosity and more carboxyl groups increase the availability of hydrophilic groups so as to easily form H-bonds with water molecules [37]. As highlighted in Figure 2B, three peaks at 3466 cm^−1^, 1645 cm^−1^, and 1085 cm^−1^, which correspond to the N-H bending, C=O stretching, and C-O stretching vibrations respectively [38], were observed in the spectrum of γ-PGA. The peak at around 1650 cm^−1^ corresponding to C-O stretching vibration (amide I band) in saturated aliphatic carboxylic acid dimers and the peaks at around 1400 cm^−1^ and 920 cm^−1^ ascribed to O-H bending vibration of carboxylic acid were also seen, confirming a large number of carboxyl groups. In the spectrum of CS, several peaks in 1510–1650 cm^−1^ attributed to N-H bending vibration [28] and the peak at around 1018 cm^−1^ corresponding to C-N stretching vibration were observed, confirming massive amino groups. In the spectrum of CS/γ-PGA hydrogel, the band shift happened at 3466 cm^−1^, 1645 cm^−1^, and 1082 cm^−1^, suggesting the occurrence of the PEC interactions between the amino groups of CS and the carboxyl groups of γ-PGA [34]. The broad peak in 3500–3000 cm^−1^ corresponding to the stretching vibration of free hydroxyl groups has been reported to be overlapped with the stretching of N-H bonds [29]. As compared with CS and γ-PGA, CS/γ-PGA hydrogel has a different peak shape at 3466 cm^−1^. This may be ascribed to the interactions of PEC and H-bond. In conclusion, the increase in γ-PGA content makes it increased not only the intermolecular space but also hydrophilic groups in hydrogel, remarkably improving water absorption and retention. This will provide more chance for DOE to induce the self-assembly of CS/γ-PGA hydrogel.

As indicated in Figure 1C, CS owns stronger antibacterial behavior than γ-PGA. With the increase of γ-PGA content, the relative content of CS in CS/γ-PGA hydrogels reduced, weakening the antibacterial behavior of the hydrogels. As well known, CS inhibits Gram-positive bacteria stronger than gram-negative bacterium, leading to *S. aureus* to be inhibited stronger than *E. coli* in all samples. *S. aureus*, a Gram-positive bacterium, owns ample teichoic acid in the hydrophilic cell wall. CS molecules interact with the acidic molecules so easily as to inhibit the synthesis of cell wall [39]. But in Gram-negative bacterium like *E. coli*, hydrophobic cell wall consisting of water-immiscible lipopolysaccharide and outer membrane proteins weaken the antibacterial effect of CS [40]. On the other hand, the outer and inner membranes of Gram negative bacteria prevent heavy molecular weight γ-PGA from entering cell so that γ-PGA molecules cover the cell surface to block mass transfer [41]. However, smaller γ-PGA can pass through the cell membrane and then may regulate DNA transcription [42]. Interestingly, CS/γ-PGA hydrogels with 14% and 10% γ-PGA have stronger inhibition than neat CS, indicating a possible synergic effect of γ-PGA with CS in enhancing the antibacterial capability. *S. aureus* lives comfortably at pH 7.4, higher than the antimicrobial critical value of CS, pH 7.0 [43]. The addition of γ-PGA can reduce pH value to inhibit the propagation of microbial cells. Besides, PEC interactions between γ-PGA and CS could further enhance the synergy effect on antibacterial capability [42]. In conclusion, 25% γ-PGA was applied to prepare CS/γ-PGA hydrogels in the next work.

### 3.2. Effect of DOE

After cultured with *Lactobacillus*, there are 131 mg/mL total sugars and 0.186 mg/mL total phenols in DOE solution. As an enzymatic fingerprinting of DOP, amylase activity was hardly changed before and after the fermentation. It can hydrolyze common β-1,4-linked backbone of DOP to produce low MW polysaccharides and monosaccharides [44], improving DOE in DPPH^+^ scavenging activity, reducing power and hydroxyl radical scavenging rate as listed in Table 1. DOE includes 0.186 mg/mL total phenols and 4.75 U/mL SOD, further enhancing its antioxidative behavior. With fermentation, protease was increasingly produced to degrade proteins into various size of peptides and amino acids with amphipathic and hydrophilic groups. They can be convinced as an ideal bridge to link the strands of biomacromolecules (Figure 1). Previously, CS and γ-PGA were directly dissolved in acetic acid solution to make rapid PEC interactions between them [45]. However, the rapid interactions are not conducive to the formation of uniform hydrogels. CS exhibits pH-dependent in the self-assembly to name a few [46,47]. It is difficult to dissolve in alkaline solution but easy to crosslink with other compounds in an acidic solution [48]. Based on this, mild PEC interaction was designed to construct an uniform CS/γ-PGA hydrogel. In detail, γ-PGA and CS were firstly fully mixed until a uniform distribution of CS in sodium hydroxide was obtained. This was followed by the dropwise addition of protonated amino groups of CS to the acetic acid solution to generate gradual PEC interactions. Be that as it may, the difference of CS from γ-PGA in the dosage also affect the self-assembly of CS/γ-PGA hydrogel, changing pore size and structural uniformity of the network.

As illustrated in Figure 1, the skeleton network of CS/γ-PGA hydrogel is not uniform. With the intrusion of DOE, amino acids with short C-chains permeate easily into smaller pores in the uneven network and act as a bridge to interlink the strands of CS and γ-PGA through PEC interactions [49]. As regarding large pores, peptides can enter and strengthen them like amino acids. As for the largest pores, DOPs with longer strands can bridge them. Some anionic DOPs can form PECs with CS and binds to γ-PGA through H-bonds. The multiple noncovalent crosslinking reactions inevitably change self-assembly of hydrogels. As highlighted in Figure 2(A-c), many layered nanostructures like circular protuberances (*Φ* 200 nm or so) were formed. It follows that amino acids, peptides and polysaccharides in DOE lead to the self-assembly of the hierarchical structure based on multiple crosslinking. As proved in Figure 2C, the addition of DOE decreased the intensity at 22.56°, decreasing crystal production orientation in this direction. Also, the shift was observed at 15.78°, in particular as 4% DOE added, the crystal phase in this direction was changed, proving the role of DOE in changing the self-assembly of CS/γ-PGA hydrogel.

FTIR spectra of neat DOE and CS/γ-PGA/DOE hydrogel in the region of 400–4000 cm^−1^ were arranged in Figure 2B. In the spectrum of DOE, the wider peak in 3800–2800 cm^−1^ (enclosed in box 1) corresponding to carboxyl groups shifted to low wavelength relative to the ones of CS and γ-PGA because of the dimerization of carboxyl groups. Another specific peak observed at 1555 cm^−1^ in 1620–1450 cm^−1^ is ascribed to benzene ring vibration [50]. This is because DOE contains a large amount of amino acids and polyphenols. As compared with CS/γ-PGA hydrogel, CS/γ-PGA/DOE hydrogel has a peak shift from 3466 cm^−1^ to a low wavelength, and the peak becomes wider. The bands in 3455–3410 cm^−1^ and 3375–3340 cm^−1^ refer to intramolecular hydrogen bond, and the band in 3310–3230 cm^−1^ refers to intermolecular hydrogen bond [51]. The band shift could result from the formation of intramolecular hydrogen bonds between DOE, γ-PGA and CS, and the increase in the amount of hydroxyl groups widen the peak. Besides, two obvious differences were observed between the peaks enclosed in box 2 and box 3, where the peaks at 1645 cm^−1^ and in 1000–1400 cm^−1^ correspond to the dimerization of saturated aliphatic carboxylic acid and out-of-plane vibration of C-H bonds, respectively, indicating DOE bringing massive amino acids and polyphenols into the skeleton network of CS/γ-PGA. As a result, the addition of DOE accompanied by multiple noncovalent crosslinking among γ-PGA, CS and various components of DOE.

Theoretically, multiple crosslinking interactions can improve the structural uniformity and the toughness of CS/γ-PGA skeleton network. As illustrated in Figure 3a,b, 4–6% DOE significantly enhanced the tensile behavior of CS/γ-PGA/DOE hydrogel. In particular, the hydrogel with 4% DOE has the stress at break of 1.58 MPa, the elongation at break of 89.4% and the Yong’s modulus of 2.07 MPa, 2.16-fold, 2.76-fold and 1.36-fold those of CS/γ-PGA hydrogel, respectively. But, too much or too little DOE cannot provide an improvement. Too much DOE leads to more multiple crosslinking interactions and competitively obstruct PEC interactions in the skeleton network of CS/γ-PGA, reducing the tensile behavior. On the contrary, too little DOE maybe results in insufficient multiple crosslinking interactions. Therefore, moderate DOE is the optimum to maintain the tensile behavior.

As arranged in Figure 2D, the addition of DOE increases weight loss below 100 °C and around 300 °C, particularly when 4% DOE was added. As reported, the degradation process of macromolecules such as polysaccharides, in general, goes through the following steps: dehydration, depolymerization and the decomposition [52,53]. The initial mass loss below 200 °C mainly results from the removal of free water [54], the loss at 275 °C–350 °C the random cleavage of glycosidic bonds [55], and the loss at 300–600 °C the decomposition of the saccharide ring structure [56]. Relative to covalent crosslinking, noncovalent crosslinking can be easier broken at around 300 °C. The increase in weight loss below 100 °C indicates more free water in the porous network, confirming the increase of the porosity of CS/γ-PGA/DOE hydrogel. 4% DOE is the optimum to improve the porosity of the hydrogel, proving the importance of moderate DOE in maintaining the equilibrium between the structure and behaviors. The macroporosity and interconnectivity of macropores in the network of hydrogels determine its free volume, affecting its water absorption and release [57]. In this work, DOE increased the porosity of CS/γ-PGA/DOE hydrogels to take in more water. Water retention depends on the amount and the species of hydrophilic units on the network of hydrogel [58]. As highlighted in Figure 3c, water absorption was higher in CS/γ-PGA/DOE hydrogels with 2–5% DOE than in CS/γ-PGA hydrogel, and in particular, the addition of 3–4% DOE made water absorption increased by approximate 39% but water retention decreased by 10.7%. It follows that CS/γ-PGA/DOE hydrogels can keep more water than CS/γ-PGA hydrogel. Also, DOE can improve CS/γ-PGA/DOE hydrogel in antibacterial and DPPH^+^ scavenging behaviors. As illustrated in Figure 3d, the inhibition zone of CS/γ-PGA/DOE hydrogel against *E. coli* and *S. aureus* increases with DOE. This is ascribed to proteases and DOPs of DOE. With fermentation, proteases of DOE hydrolyze various proteins in the wall of the bacteria, and various DOPs interact with the proteins, inhibiting the growth of *E. coli* and *S. aureus* [59].

On the other hand, DOE contains many DOPs and SOD [60] that remove various oxygen free radicals, improving significantly antioxidative property of γ-PGA/CS/DOE hydrogel. The antioxidative effect of DOE was investigated by evaluating the DPPH^+^ scavenging ability. As highlighted in Figure 3e, DPPH^+^ scavenging behavior increases with DOE. Reasonably, various antioxidative DOPs can reduce the generation of reactive oxygen species and oxidative free radicals [25,26], and microbial fermentation further enhances the reduction. In vitro cytotoxicity assay only intended to provide preliminary evaluation of CS/γ-PGA/DOE hydrogel for potential skin care. The cytocompatibility was examined with HEK-293 cell. As highlighted in Figure 3f, the cell viabilities on the hydrogel with 3–5% DOE are almost over 95% as compared with CS/γ-PGA hydrogel, far more than 70% that is a common indicator for cytocompatibility. Interestingly, low concentration of DOE such as 3% can promote cell proliferation, revealing the good cytocompatibility of CS/γ-PGA/DOE hydrogel. As well known, CS and γ-PGA are non-toxic natural polymers, and CS/γ-PGA PEC hydrogels with better biocompatibility have been used as wound dressing material [42,61]. DOP is not a toxin but an effective therapeutic reagent to attenuate secondary liver injury [62]. In vitro fermented DOPs have prebiotic potential and may improve gastrointestinal health [63]. Especially, multiple noncovalent crosslinking interactions among CS, γ-PGA, and DOE are reversible and no toxic side effects. Therefore, CS/γ-PGA/DOE hydrogel has real cytocompatibility to human body.

### 3.3. Characterization of CS/γ-PGA/DOE Hydrogel for Potential Skin Care

Nowadays, skin texture to a desired smooth and young appearance is attracting more and more interest. Skin care products have many requirements such as moisturizing, antioxidant, air permeability, and other nourishing function [64]. DOE improved CS/γ-PGA/DOE hydrogel in antioxidative behavior, water absorption, water retention, hydrophilicity and antibacterial behavior, which contribute to skin care. Besides, massive hydrophilic groups of DOE ingredients can improve the wettability of CS/γ-PGA/DOE hydrogel. As depicted in Figure 4a, the contact angle and rolling angle of CS/γ-PGA hydrogel are 86.74° and 25°, respectively, and those of CS/γ-PGA/DOE hydrogel 68.72° and 44.23°. They are both less than 70°, indicating a good hydrophilicity [65]. DOE can reduce the contact angle of CS/γ-PGA/DOE hydrogel but increase the rolling angle, manifesting the wettability of the hydrogel being significantly improved.

Air permeability is another important comfort index of skin care products. The oxygen permeability was investigated to characterize air permeability of the hydrogel. As shown in Figure 4b, DOE significantly increased the oxygen permeability of CS/γ-PGA/DOE hydrogel by 52%. Various size of molecules in DOE could induce skeleton network of CS/γ-PGA to form a more regular 3D network, promoting air permeability [66]. Furthermore, many skin-nourishing ingredients of DOE would have played roles in skin care if they were released from CS/γ-PGA/DOE hydrogel in the use. The releasing rate of DOE from the hydrogel is great important for skin-nourishing. As illustrated in Figure 4c, the DOE in the hydrogels dried by heating and freezing both had been released to the balance in 0.5 h, approximate use time of skin-care products. In this period, the heating-dried hydrogel released 51% DOE, the freezing-dried one 85% DOE. This is ascribed to that freezing forms a great number of ice crystals to increase the porosity of the hydrogel [35] (Figure 2A). As a result, water can easily exchange with various size of DOE ingredients noncovalently crosslinked to the skeleton network of CS/γ-PGA. The released ingredients should directly attach to the surface of skin for better care if a wet CS/γ-PGA/DOE hydrogel were to be covered on skin. Therefore, the hydrogel will be also potentially extended in the administration of transdermal delivery, which has proven to be one of the most favorable methods among novel drug delivery systems [67].

The pH value of human beings’ skin, in general, ranges from 4.0–7.0 [68]. Higher pH value often leads to skin aging, and lower one can enhance skin sensitivity [69]. CS/γ-PGA/DOE hydrogel has a good pH adaptability to skin. As indicated in Figure 5, the surface of CS/γ-PGA/DOE hydrogel exhibits a mild acidity (pH 6.0–6.5) suitable for human beings’ skin. On the other hand, the hydrogel owns a good regeneration behavior. The dry hydrogel before use is white and wrinkled, and it becomes transparent and smooth as soon as being wetted. The moist hydrogel had been compatibly attached to skin for more than 30 min and still been transparent, exhibiting a attracting water retention of the hydrogel. After used for 30–60 min, the hydrogel didn’t bring no uncomfortable feeling, and the skin looked cleaner and more sheeny. With time, the hydrogel will become dry, white, and stiff. Be that as it may, upon incubated in distilled water for less than 10 min, the used hydrogel can rapidly return to the original state, indicating a good regeneration for potential repeated use.

## 4. Conclusions

In summary, DOE leads to multiple noncovalent crosslinking interactions with the skeleton network of CS/γ-PGA, significantly changing the self-assembly of CS/γ-PGA/DOE hydrogel and improving the porosity and the toughness. It is convinced that multiple noncovalent crosslinking based on DOE can effectively stabilize the network structure of hydrogels and improve the toughness. The new-formed protuberance nanostructures exhibits better water absorption and air permeability, potential for skin care. On the other hand, the intrusion of many skin-nourishing ingredients of DOE also enhances the hydrogel in the behaviors of antioxygenation, antibacterial and hydrophilicity, meeting the requirements for skin care products. Particularly, the enhanced toughness makes CS/γ-PGA/DOE hydrogel not be easily broken as applied in skin care. The hydrogel is prepared completely with natural compounds biocompatible to human body. It has been proved to be healthy for skin care without side action. Totally, this work promotes hydrogels in practical applications.

## Data Availability

Not applicable.

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
