# Peer review of "Dendrobium officinale Enzyme Changing the Structure and Behaviors of Chitosan/γ-poly(glutamic acid) Hydrogel for Potential Skin Care"

_polymers, 2022, doi:10.3390/polym14102070_

Round 1

Reviewer 1 Report

This paper reports dendrobium officinale enzyme changing the structure and behaviors of chitosan/γ-poly(glutamic acid) hydrogel. DOE leads to multiple noncovalent crosslinking interactions with the skeleton network of CS/γ-PGA, significantly changing self-assembly of CS/γ-PGA/DOE and improving the porosity and the toughness. The result and discussion are interesting. However, the points in the manuscript still need some improvement. The specific comments are as follows:

  1. All Figures should have good resolution.
  2. “In consideration of the correlationship between the structure and the behaviors of the adsorption, water absorption and water retention, 25 % γ-PGA was applied to prepare CS/γ-PGA hydrogels in the next work” The authors should state the reason, why?
  3. Protease may have cytotoxicity on cell. The mass concentration of CS/γ-PGA/DOE hydrogel should be offered in Figure 3f.
  4. Effect of DOE concentration on tensile behaviors was confusing, please check the results.
  5. DOE was complex mixture; the authors should state the method of determining the concentration of DOE. DOE release behavior is in Figure 4C, which component of DOE was measured?
  6. Please carefully check the text for writing and grammar.

Author Response

Reviewer #1:

This paper reports dendrobium officinale enzyme changing the structure and behaviors of chitosan/γ-poly(glutamic acid) hydrogel. DOE leads to multiple noncovalent crosslinking interactions with the skeleton network of CS/γ-PGA, significantly changing self-assembly of CS/γ-PGA/DOE and improving the porosity and the toughness. The result and discussion are interesting. However, the points in the manuscript still need some improvement. The specific comments are as follows:

  1. All Figures should have good resolution.

Response: Fig.1 was modified and other figures were made clearer.

  1. “In consideration of the correlationship between the structure and the behaviors of the adsorption, water absorption and water retention, 25 % γ-PGA was applied to prepare CS/γ-PGA hydrogels in the next work” The authors should state the reason, why?

Response: Thanks for your question. This part has been revised.

  1. Protease may have cytotoxicity on cell. The mass concentration of CS/γ-PGA/DOE hydrogel should be offered in Figure 3f.

Response: You are right. Protease has detrimental effects of cell wall. But in this work, proteases didn’t solely exist and play roles in skin with other ingredients of DOE including polysaccharides, which can protect cell to a certain. MTT assay also proved it. The mass concentration of CS/γ-PGA/DOE hydrogel was designed as 15% (μg/μL) in Figure 3f.

  1. Effect of DOE concentration on tensile behaviors was confusing, please check the results.

Response: Thanks for your question. This part has been rewritten in the revised version.

  1. DOE was complex mixture; the authors should state the method of determining the concentration of DOE. DOE release behavior is in Figure 4C, which component of DOE was measured?

Response: Thanks. This is a good question. DOE is a mixture composed of various ingredients and thus it is difficult to determine an accurate concentration of DOE. However, DOE includes plentiful phenols and thus it has the characteristic peak at 270 nm. Therefore, to rough investigate the concentration using UV spectrum at 270 is feasible. This part has supplemented in the revise MS.

  1. Please carefully check the text for writing and grammar.

Response: Thanks for your careful check. I have throughout checked and modified the MS.

Reviewer 2 Report

  • Introduction, line 30: please write the full form of CS and PGA first.
  • Line 30-31, the sentence needs a reference.
  • This sentence needs to rewrite because it has grammatical errors: “CS/γ-PGA hydrogel is important one of many traditional hydrogels.” it must be written like this: CS/γ-PGA hydrogel is one of many important traditional hydrogels.
  • From lines 72 to 82, there are no references. After each fact, a proper reference must be cited.
  • In the Materials and Methods section: DPPH is an abbreviation. First, write its full form
  • In the last line of the chemicals section, the information given about the strains is not acceptable and the company for the supplier must be mentioned.
  • There is no reference after each methodology such as Preparation of DOE, Preparation of precursor solution of hydrogel, Preparation of hydrogel,..
  • The preparation of hydrogel is incomplete. The volume of the water, heating temperature, time of stirring, time of polymerization, and … must be mentioned clearly.
  • The authors can use the following reference in their study:

Sabbagh, F., & Kim, B. S. (2022). Recent advances in polymeric transdermal drug delivery systems. Journal of Controlled Release341, 132-146.

Author Response

Reviewer #2:

1.Introduction, line 30: please write the full form of CS and PGA first.

Response: Thank you for your careful check. We added the full name as you said.

  1. Line 30-31, the sentence needs a reference.

Response: Thank you for your careful check. Some references were cited for the sentence.

  1. This sentence needs to rewrite because it has grammatical errors: “CS/γ-PGA hydrogel is important one of many traditional hydrogels.” it must be written like this: CS/γ-PGA hydrogel is one of many important traditional hydrogels.

Response: Thank you for your suggestion. The revision was done according to the suggestion.

  1. From lines 72 to 82, there are no references. After each fact, a proper reference must be cited.

Response: Thank you for your suggestion. The relevant references have been cited in the corresponding part.

  1. In the Materials and Methods section: DPPH is an abbreviation. First, write its full form

Response: Thank you for your careful check. The full form was supplemented in the revised MS.

  1. In the last line of the chemicals section, the information given about the strains is not acceptable and the company for the supplier must be mentioned.

Response: Thank you for your suggestion. The information about strains have been added in the revised version.

  1. There is no reference after each methodology such as Preparation of DOE, Preparation of precursor solution of hydrogel, Preparation of hydrogel,..

Response: Thank you for your suggestion. The relevant references have been cited and the methodology has been detailed.

  1. The preparation of hydrogel is incomplete. The volume of the water, heating temperature, time of stirring, time of polymerization, and … must be mentioned clearly.

Response: Thank you for your suggestion. The detail protocol was present.

  1. The authors can use the following reference in their study:

Sabbagh, F., & Kim, B. S. (2022). Journal of Controlled Release, 341, 132-146.

Response: Thank you for your suggestion. This valuable reference was cited in the revised MS.
